# A Review of the Potential Application of Osteocyte-Related Biomarkers, Fibroblast Growth Factor-23, Sclerostin, and Dickkopf-1 in Predicting Osteoporosis and Fractures

**DOI:** 10.3390/diagnostics10030145

**Published:** 2020-03-06

**Authors:** Fitri Fareez Ramli, Kok-Yong Chin

**Affiliations:** Department of Pharmacology, Faculty of Medicine, Universiti Kebangsaan Malaysia, 56000 Cheras, Malaysia; fitrifareez@ppukm.ukm.edu.my

**Keywords:** bone, bone turnover, skeleton, screening, Wnt signaling pathway

## Abstract

Bone turnover markers (BTMs) derived from the secretory activities of osteoblasts and the matrix-degrading activities of osteoclasts are useful in monitoring the progression of osteoporosis and the efficacy of anti-osteoporotic treatment. However, the usefulness of BTMs in predicting osteoporosis remains elusive. Osteocytes play a central role in regulating bone formation and resorption. The proteins secreted by osteocytes, such as fibroblast growth factor-23 (FGF23), sclerostin (SOST), and dickkopf-1 (DKK1), could be candidates for osteoporosis screening and fracture prediction. This review summarizes the current evidence on the potential of osteocyte-related proteins as biomarkers for osteoporosis and fracture prediction. The literature reports that SOST may be a potential marker for osteoporosis screening but not for fracture prediction. FGF23 is a potential marker for increased fracture risk, but more studies are needed to confirm its usefulness. The role of DKK1 as a marker to predict osteoporosis and fracture risk cannot be confirmed due to a lack of consistent evidence. In conclusion, circulating osteocyte markers are potential osteoporosis biomarkers, but more studies are warranted to validate their clinical use.

## 1. Introduction

Bone turnover markers (BTMs) are derived from the secretory proteins of osteoblasts (bone formation markers) and the matrix degradation products of osteoclasts (bone resorption markers). They provide a snapshot of the bone remodelling process, which is valuable in monitoring the progression of bone-related diseases and the efficacy of osteoporosis treatment [1,2]. However, traditional BTMs may not represent individual skeletal health status because multiple studies have reported the absence of a significant association between BTMs and bone mineral density (BMD) [3], even among patients with osteoporosis [4]. In addition, multiple studies show conflicting results in the use of BTMs for fracture risk assessment [5,6,7].

Recent studies have unveiled the role of osteocytes, the most abundant cells in the skeleton, in regulating bone turnover [8]. Osteocytes are capable of remodelling the perilacunar matrix in calcium-demanding situations like lactation [9]. Osteocytes also act as mechanosensors in the bone and regulate the activities of osteoblasts and osteoclasts [10]. Upon mechanical stimulation, osteocytes generate nitric oxide and prostaglandin to promote bone formation. On the other hand, they secrete sclerostin (SOST) and dickkopf-1 (DKK1) to inhibit bone formation [11,12]. Osteocytes also secrete osteoprotegerin and receptor activator of nuclear factor kappa-Β ligand (RANKL) to regulate osteoclast differentiation [10,13,14,15]. Osteocytes also produce fibroblast growth factor-23 (FGF23) in their regulation of phosphate homeostasis [16].

In view of the specific role of osteocyte-related proteins such as DKK-1, SOST, and FGF23 in regulating bone health, it is tempting to speculate that their circulating levels could predict BMD, osteoporosis, and fracture. This review aims to discuss the potential utilization of these bone markers in predicting individual bone health status and fracture risk.

## 2. Fibroblast Growth Factor-23

FGF23 is mainly produced by osteocytes and is an important regulator of phosphate and 1,25-dihydroxyvitamin D3. Phosphate plays an important role in bone mineralization, and most of the phosphate in the body can be found in bone [17]. The kidney is one of the main target organs for FGF23, wherein FGF23 as well as other osteocyte-derived factors such as acidic serine–aspartate-rich matrix extracellular phosphoglycoprotein (MEPE)-associated motif (ASARM) peptides regulate phosphate in the kidney by reducing phosphate reabsorption in the proximal tubules via a sodium phosphate cotransporter [18,19]. Moreover, FGF23 can inhibit the expression of renal 1 hydroxylase, which converts 25-hydroxyvitamin D into 1,25-dihydroxyvitamin D3 (the active form), subsequently interfering with calcium homeostasis [20]. An excessive FGF23 level is associated with hypophosphatemia and manifests as rickets in children and osteomalacia in adults [17]. The relationship between FGF23 and bone mineralization may be similar in other bone diseases like osteoporosis. 

The relationship between FGF23 and BMD has been explored by several observational studies. Coulson et al. [21] reported no significant association between FGF23 and whole-body BMD in the subjects of their study, regardless of age. The level of FGF23 was similar between young (18–30 years) and old (69–81 years) subjects in their study [21]. A prospective study conducted in the elderly population aged between 70–79 years in the US also showed no significant association between FGF23 levels and the annual percentage changes in total hip areal BMD after adjusting for demographics, BMI, and estimated glomerular filtration rate (eGFR) [22]. 

Some sex-specific studies on the link between FGF23 and BMD have been reported. Among elderly men in Sweden, no significant association between FGF23 levels and BMD at the hip and lumbar spine was found [23]. Among Caucasian and African American premenopausal women, negligible associations between FGF23 and both spine and femoral neck BMD were reported [24]. A study among Chinese postmenopausal women also reported no significant correlation between FGF23 and lumbar BMD [25]. In contrast, Celik et al. [26] found that postmenopausal women with osteoporosis had significantly higher FGF23 levels compared to postmenopausal women with osteopenia and healthy controls. 

BMD is a gross measure of bone health status and does not reflect the microarchitecture of the bone [27]. Rupp et al. [4] found that high FGF23 levels were associated with impaired trabecular microarchitecture at the distal radius and tibia defined by bone volume, trabecular thickness, and number among subjects with osteoporosis. However, the association between FGF23 and cortical aspects (thickness and BMD) was not significant. In addition, they found no significant association between FGF23 serum levels and BMD at both hip and spine [4]. Thus, FGF23 may be linked to minute changes in trabecular bone microarchitecture not detected by BMD. 

Despite the lack of a relationship between FGF23 and BMD, some studies reported a significant association between FGF23 levels and the risk of an osteoporotic fracture. Mirza et al. [28] reported a significant association between higher FGF23 levels and an increased risk of vertebral fractures but not non-vertebral fractures among 2868 Swedish elderly men. The fracture risks were the greatest when FGF23 levels were more than 55.7 pg/mL [28]. Similarly, a study in the Japanese population with early chronic kidney disease (CKD) reported that an FGF23 level of 56.8 pg/mL was the optimal cut-off point for vertebral fracture prediction [29]. 

In contrast, Lane et al. [30] did not find any significant association between the incidence of spine, non-spine, major osteoporotic fractures or vertebral fractures and FGF23 among Swedish elderly men. Isakova et al. [22] and Jovanovich et al. [31] also reported no significant association between fracture risk and FGF23 levels among the elderly population, regardless of sex. 

Stratification based on estimated glomerular filtration rate has a significant impact on the association between FGF23 and fracture risk. Mirza et al. [28] found a 31% increase in the risk of all fractures in patients with eGFR > 71.5 mL/min/1.73 m^2^, but no significant association was found in patients with eGFR below this value. In contrast, Lane et al. [30] found a two-fold increased risk of non-vertebral fractures in patients with eGFR < 60 mL/min/1.73 m^2^ in elderly men, but no significant association was found between FGF23 levels and fracture risk above this value. 

## 3. Sclerostin

SOST is a bone metabolism regulator in the canonical Wnt signalling pathway. It is exclusively produced by mature osteocytes or late-stage osteoblasts. It exerts an inhibitory effect on osteoblast activity by binding to lipoprotein receptor-related protein (LRP) 5 and 6 [32]. Preclinical studies have shown that targeted SOST antibody treatment results in increased bone mass [33]. This observation has been translated into clinical studies, where postmenopausal women with osteoporosis receiving SOST antibody treatment demonstrated a significant positive effect on BMD [34].

In a population study that included both sexes, a significant positive association between SOST and BMD was reported in older subjects but not in younger subjects after adjusting for confounding factors [21]. Significant age and sex differences in SOST levels were observed, whereby the SOST level was significantly higher in older subjects and men compared to younger subjects and women [21]. 

Other studies reported a significant positive association between serum SOST levels and BMD in postmenopausal women. Postmenopausal women with osteoporosis were reported to have a lower SOST level compared to postmenopausal women without osteoporosis [35]. A positive association between SOST levels and BMD at the lumbar spine [35,36,37,38,39], femoral neck [35,36,37], trochanter [37], total hip [35,36,37,40], and whole body [35,41] was observed in postmenopausal women. In men, a study reported that serum SOST levels were positively associated with BMD at the spine, hip, and whole body among subjects aged 20–87 years [42].

In contrast, a study in a Turkish population found a significant negative association between SOST levels and spine and femoral neck BMD [43]. Studies in both Chinese [44] and Spanish [45] women reported that postmenopausal women with osteoporosis had a lower SOST level compared to postmenopausal women without osteoporosis, but a study in Indonesia [46] reported otherwise. 

In terms of risk for osteoporotic fracture, the current literature reveals heterogeneous results. Lim et al. [37] reported that postmenopausal Korean women had a > 1.5-fold increased risk of fractures for every one standard deviation decrement of SOST levels. In contrast, a study conducted among postmenopausal women in Jeddah found an approximately eight-fold increase in fracture risk for every one standard deviation increase in SOST levels [41]. The fracture-predictive value of SOST started from year one following the first visit and remained significant for five years of follow-up. The cut-off points of ≥68.88 pmol/L showed the highest attributable risk of osteoporotic fracture [41]. Similarly, Arasu et al. [40] found a 50% increased risk of hip fracture with each 1 SD increment among Caucasian women aged ≥ 65 years old. In men, Szulc et al. [42] found a positive association between serum SOST level and osteoporotic fracture.

Other studies have reported a negligible association between SOST and osteoporotic fracture. Garnero et al. [47] found no significant association between serum SOST levels and risk of all fractures in a prospective study involving 572 postmenopausal women. Luque-Fernandez et al. [45] reported no significant difference in serum SOST levels between Spanish postmenopausal women with and without an osteoporotic fracture. 

## 4. Dickkopf-1

DKK1 is another antagonist in the canonical Wnt pathway. Similar to SOST, DKK1 has a similar affinity towards LRP5 and LRP6. Moreover, DKK1 also binds to the Kremen2 receptor involved in the Wnt signalling pathway [48]. DKK1 antibodies prevent the binding of DKK1 to LRP6 and Kremen2 receptors, resulting in increased bone formation and BMD of animals in preclinical studies [48]. 

A recent animal study found that systemic levels of DKK1 mainly originate from osteoprogenitors and not the late osteoblasts or osteocytes [49]. However, DKK1 expression by early and late osteoblasts as well as osteocytes is crucial for local bone modulation [49]. Other cell types such as chondrocytes and adipocytes in bone marrow and gastric epithelium may also contribute to systemic DKK1 levels [49,50,51]. Thus, despite its prominent role in regulating bone mass, systemic DKK1 levels may not truly reflect osteocytic activity. 

A multicenter osteoporosis study conducted in five European countries reported a significant positive association between DKK1 and whole-body BMD in subjects, regardless of age and sex [21]. Tian et al. [44] found a significant negative correlation between serum DKK1 levels and spine BMD in 350 Chinese postmenopausal women with osteoporosis after adjusting for confounding factors. Similarly, other studies reported a negative correlation between DKK1 levels and BMD at the femoral neck region [52,53]. The levels of DKK1 were significantly higher in postmenopausal women with osteoporosis compared to healthy controls [44,53]. In a study by Butler et al. [52] involving 36 subjects, a higher level of DKK1 was observed in patients with osteoporosis compared with healthy controls. In contrast, a study in Korean postmenopausal women reported no significant association between DKK1 and hip and spine BMD [37]. 

In terms of osteoporotic fracture risk, Lim et al. [37] found no significant association with DKK1 levels between postmenopausal women with or without osteoporotic fractures. On the other hand, Dovjak et al. [54] reported higher DKK1 levels in male patients with hip fractures compared to the healthy control group.

A summary of the relationship between osteocytes biomarkers discussed above and BMD or fracture risk is summarized in Table 1. 

## 5. Conclusion and Perspectives

The evidence thus far supports SOST as a predictor of osteoporosis but not osteoporotic fractures, while FGF23 is a predictor of osteoporotic fractures, depending on more validation studies. The role of DKK1 as a predictive tool for osteoporosis and fractures requires further study because the available evidence is limited. Despite the existing evidence, these markers should not replace dual-energy X-ray absorptiometry as the gold standard technique for diagnosing osteoporosis. They may help to provide insights on the underlying causes of low BMD or fracture experienced by patients. In addition, they should be used together with other BTMs because bone remodelling is a complex process involving multiple cell types. Since the reference values for these osteocyte-related markers are not established, their diagnostic values could be questionable. Therefore, baseline values for individual patients should be obtained and monitored throughout the disease and treatment process to make a logical judgement of the bone health status of the patients. The physicians should also interpret the level of these biomarkers together with factors influencing their levels, such as age, sex, eGFR, and other variables related to calcium homeostasis. 

## Figures and Tables

**Table 1 diagnostics-10-00145-t001:** The relationship between osteocyte markers and bone mineral density or osteoporotic fractures.

Ref	Study Design	Sample Population	Association with BMD			Difference between Groups			Association with OF			Difference between OF and Non-OF		
			FGF23	SOST	DKK1	FGF23	SOST	DKK1	FGF23	SOST	DKK1	FGF23	SOST	DKK1
[4]	Cross-sectional study	82 German patients with osteoporosis (66 women and 16 men). Mean age was 64 years.		Nil ^a,d^										
[21]	Cross-sectional study	272 older men and women (69–81 years) and 171 younger men and women (18–30 years) in five European countries.	Nil	+ in old group ^e^	+ in old, young, all ^e^	Nil (old vs. young) and male (male vs. female)	↑ in old (old vs. young) and male (male vs. female)	↑ in old (old vs. young)						
[22]	Prospective cohort study	2786 elderly population, Caucasian and African American men and women, aged 70–79 years.	Nil ^d^						Nil					
[24]	Cross-sectional study	1631 Caucasian and 296 African American women, aged 20–55 years.	Nil ^a,b^											
[25]	Cross-sectional study	355 postmenopausal Chinese women (aged 62.92 + 8.78 years).	Nil ^a^			Nil *								
[26]	Case–control	28 Turkish postmenopausal osteoporosis (PMO), 32 postmenopausal osteopenia, 30 controls.	BMD ↓ in PMO			↑ In PMO (compare to osteopenian and controls)								
[28]	Prospective cohort study	2868 Swedish older men (75.4 ± 3.2 years).							↑ OF ^§^					
[30]	Prospective cohort study	1772 older men in the US, 90% Caucasian, mean age 73 years.							↑ in OF ^†^					
[31]	Prospective cohort study	2008 women and 1329 men, Caucasian and African American (16%), older population, aged 78 ± 5 years.	+ in men ^a,d^			↓ in men (men vs. women)			Nil (in men or women)					
[35]	Cross-sectional study	260 postmenopausal Chinese women (50–76 years).		+ ^a,b,d,e^			↓ in PMO (PMO vs. PMNO)							
[36]	Cross-sectional study	703 Chinese postmenopausal women (50–94.5 years).		+ ^a,b,d^										
[37]	Case–control study	103 osteoporotic fracture cases and 103 controls of postmenopausal women (62.8 + 6.1 years in each group).		+ ^a,b,c,d^	Nil ^a,b,c,d^					↑ in OF as SOST ↓	Nil		↓ in OF ^§^	Nil
[38]	Interventional study	49 Greek Postmenopausal women (50–80 years).		+ ^a^										
[39]	Cross-sectional study	352 postmenopausal Japanese women (65.5 + 9.3 years).		+ ^a^										
[40]	Prospective case–control study	Caucasian women aged at least 65 years old (228 hip fractures, 204 controls).		+ ^d^			Nil (case vs. control)			↑ in OF				
[41]	Prospective cohort study	707 Arabic postmenopausal women (53–91 years).		+ ^e^									↑ in OF	
[42]	Cross-sectional study	1134 French men (20–87 years).		+ ^a,d,e^									↓ in OF	
[43]	Cross-sectional study	135 Turkish postmenopausal women (68.1 + 9.5 years).		− ^a,b^										
[44]	Case–control study	500 (350 postmenopausal Chinese women with osteoporosis and 150 controls), mean age 58.6 years.			− in PMO ^a^		↓ in PMO (PMO vs. control)	↑ in PMO (PMO vs. control)						
[45]	Cross-sectional study	97 postmenopausal Spanish women (41–83 years).		Nil			↓ in PMO (PMO vs. PMNO)						Nil	
[46]	Case–control study	120 Indonesian postmenopausal women (60 with osteoporosis and 60 without), aged between 55–70 years.		Nil in PMO and PMNO			↑ in PMO (PMO vs. PMNO)							
[47]	Prospective cohort study	572 French postmenopausal women (67 + 8.5 years).		+ ^a,d^									Nil	
[52]	Case–control study	18 patients and 18 controls, Irish men and women (48.2 + 19.8 years).			− ^a,b^			↑ in OP(OP vs. control)						
[53]	Case–control study	88 (44 patients and 44 controls) Iranian postmenopausal women (mean age, 59 years).			− ^b^			↑ in PMO(PMO vs. control)						
[54]	Case–control study	256 Austrian older patients (≥50 years) and 67 young controls (<50 years), men and women.					↑ in men (men vs. women)	↓ in men, ↑ older					↓ in OF ^‡^	↑ in OF ^‡^
[55]	Cross-sectional	362 women and 318 men, mainly Caucasian (98%), aged 21–97 years.		+ ^a,***^										

BMD: bone mineral density; DKK1: dickkopf-1; FGF23: fibroblast growth factor-23; OF: osteoporotic fracture; PMO: postmenopausal women with osteoporosis; PMNO: postmenopausal women without osteoporosis; SOST: sclerostin; a: lumbar spine, b: femoral neck, c: trochanter, d: total hip, e: whole body, nil: no significant association. * compared to normal or osteopenia. ↑ Higher compared to other group. ↓ Lower compared to other group. § Overall, vertebral and non-vertebral and hip fracture. † Non-vertebral fracture in patients with eGFR < 60 mL/min/1.73 m^2^. ‡ Compared to the older group. *** Middle-aged women, elderly women, middle-aged men, and elderly men.

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
