# Peer review of "A Review of the Potential Application of Osteocyte-Related Biomarkers, Fibroblast Growth Factor-23, Sclerostin, and Dickkopf-1 in Predicting Osteoporosis and Fractures"

_diagnostics, 2020, doi:10.3390/diagnostics10030145_

Round 1

Reviewer 1 Report

This review by Ramli and Chin aims to discuss the potential utilization of osteocyte-derived proteins in predicting individual bone health status and fracture risk. I agree with the authors about the scientific interest to investigate the roles of osteocyte-derived factors in clinical utilization.

However, I have some comments to the authors:

Specific Comments:

Minor points

  1. Line43-44: “Osteocytes secrete RANKL to regulate osteoclast differentiation.” Authors should cite “Nat Med 2011, 17, 1231-4” and “Nat Med 2011, 17, 1235-41”.

  1. Abbreviations: You should make the abbreviation list at the end of this review.

  1. References: Cited journals should be abbreviated according to ISO 4 rules.

Major points

  1. Line50: Authors should mention not only FGF23 but also FGF23 regulating osteocyte-derived humoral factors- such as matrix extracellular phosphoglycoprotein (MEPE) and minhibin (Cell Biochem Funct 2012, 30, 355-75).

  1. Line138: Recent study reported that DKK1 is produced in immature osteoblasts rather than in mature osteoblasts or osteocytes in bone tissues (Int. J. Mol. Sci 2019, 20, 5525; https://doi.org/10.3390/ijms20225525). In this review, DKK1 is one of the osteocyte-derived proteins. Which cells are important and do produce DKK1? Please discuss about it.

I hope my comments are helpful for achieving a better version of this manuscript,

Best of luck

Your reviewer

Author Response

Thank you for reviewing our manuscript. We value your constructive comments and they are responded in the attached response sheet. 

Reviewer 2 Report

The review reports an overview of the possible bone turnover markers useful in monitoring the progression of osteoporosis and the efficacy of anti-osteoporotic treatment.

In particular, the authors put in evidence that SOST is predictive of osteoporosis such as they speculate that DKK1 can be also considered as a predictive tool for this pathological state, while for FGF23 more validation studies are needed. 

Overall I think that this is an interesting topic but that suold be implemented.

Author Response

(The authors gave the same response as above.)
